# Impact of the COVID-19 Pandemic on the Epidemiological Situation of Pulmonary Tuberculosis–Using Natural Language Processing

**DOI:** 10.3390/jpm13121629

**Published:** 2023-11-22

**Authors:** Diego Morena, Carolina Campos, María Castillo, Miguel Alonso, María Benavent, José Luis Izquierdo

**Affiliations:** 1Servicio de Neumología, Hospital Universitario de Guadalajara, 19002 Guadalajara, Spain; ccampos.bm@gmail.com (C.C.); mariacastillogarcia37@gmail.com (M.C.); alonsomiguel23@gmail.com (M.A.); jlizquierdoa@gmail.com (J.L.I.); 2Programa de Doctorado en Ciencias de la Salud, Universidad de Alcalá, 28801 Madrid, Spain; 3SAVANA, Medsavana S.L., 28013 Madrid, Spain; mbenavent@savanamed.com; 4Departamento de Medicina y Especialidades Médicas, Universidad de Alcalá, 28801 Madrid, Spain

**Keywords:** pulmonary tuberculosis, COVID-19, artificial intelligence

## Abstract

Background: We aimed to analyze the impact of the COVID-19 pandemic on pulmonary tuberculosis (TB) using artificial intelligence. To do so, we compared the real-life situation during the pandemic with the pre-2020 situation. Methods: This non-interventional, retrospective, observational study applied natural language processing to the electronic health records of the Castilla-La Mancha region of Spain. The analysis was conducted from January 2015 to December 2020. Results: A total of 2592 patients were diagnosed with pulmonary tuberculosis; 64.6% were males, and the mean age was 53.5 years (95%CI 53.0–54.0). In 2020, pulmonary tuberculosis diagnoses dropped by 28% compared to 2019. In total, 62 (14.2%) patients were diagnosed with COVID-19 and pulmonary tuberculosis coinfection in 2020, with a mean age of 52.3 years (95%CI 48.3–56.2). The main symptoms in these patients were dyspnea (27.4%) and cough (35.5%), although their comorbidities were no greater than patients with isolated TB. The female sex was more frequently affected, representing 53.4% of this patient subgroup. Conclusions: During the first year of the COVID-19 pandemic, a decrease was observed in the incidence of pulmonary tuberculosis. Women presented a significantly higher risk for pulmonary tuberculosis and COVID-19 coinfection, although the symptoms were not more severe than patients diagnosed with pulmonary tuberculosis alone.

## 1. Introduction

Infections that cause respiratory tract diseases continue to be the cause of the greatest morbidity and mortality from infectious diseases in the world. In 2019, just three pathogens featured on the WHO Blueprint priority list for research and development. These were severe acute respiratory syndrome coronavirus (SARS-CoV), Middle East respiratory syndrome coronavirus (MERS-CoV), and Mycobacterium tuberculosis. In 2020, SARS-CoV2 was included [1].

Since then, COVID-19 has been the direct cause of hundreds of thousands of deaths worldwide over the last 3 years. During the first months of the COVID-19 pandemic, almost all countries in the world experienced a devastating impact. As the virus spread rapidly, health systems faced unprecedented pressure. Confinement measures were implemented in several countries to slow the spread, affecting the daily lives of the population. The situation generated international collaboration in search of solutions, evidencing the need for a coordinated global response to public health emergencies. The acute respiratory syndrome associated with the SARSCoV-2 virus has caused serious distortions in healthcare systems, surpassing the HIV situation of 40 years ago [1].

The pandemic has had countless economic, social, and healthcare-related effects, both direct and indirect. Spain was one of the first European countries to be affected by COVID-19 and, along with Italy, also one of the first to implement confinement as a control measure [1].

The impact of COVID-19 on hospital services was dramatic, causing personnel and financial resources to be diverted. Thus, the ability to correctly diagnose and control other pathologies was limited. In fact, the burden on health services caused by the COVID-19 emergency has led to several changes in the ordinary management of both communicable and non-communicable diseases, following the reduction in or suspension of non-urgent outpatient care [1,2].

The World Health Organization has reported that certain pathologies, such as pulmonary tuberculosis (TB), may have suffered a delay in diagnosis and the start of treatment [2].

Pulmonary TB is one of the leading causes of death and disease in many countries around the world. Mycobacterium tuberculosis is the second deadliest pathogen after the virus that causes COVID-19 [2]. Transmission of pulmonary tuberculosis occurs through the inhalation of saliva droplets or respiratory secretions from infected individuals. Once inhaled, the bacteria can establish themselves in the lungs and trigger an immune response, forming granulomas that encapsulate the bacteria. However, these structures can also act as reservoirs of infection, making it difficult to completely eradicate the bacteria. Symptoms of pulmonary tuberculosis include persistent fever, weight loss, and fatigue. Early diagnosis is crucial to prevent the spread of the disease. In 2018, countries attending the United Nations high-level meeting on tuberculosis committed to intensify their efforts to achieve the ambitious goals of treating an additional 40 million people with tuberculosis and providing prophylaxis to at least 30 million people who are at risk of contracting the disease until 2030 [3]. It is considered a true healthcare problem, reaching a prevalence of 10 million patients and 1.5 million deaths in 2020, the majority registered in Asia (55%) and Africa (30%) [2,3]. The number of deaths increased from 2019 to 2020, reversing the trend of previous years [3,4,5,6]. These data are far from reaching the goal of eradicating TB in 2030 [4].

Around the world, there have been reports that some 18% of patients with pulmonary TB were not diagnosed in 2020, as the number of notified cases dropped from 7.1 million subjects in 2019 to 5.8 million in 2020 [2,3,4,5,7,8,9,10]. Although 6.4 million cases were diagnosed in 2021, we are still far below pre-pandemic levels. In Europe, a total of 47 of 504 cases of tuberculosis were reported in 2019, which is equivalent to a rate of 9.2 cases per 100,000. In 2020, however, 33,000 cases were reported, most of whom were male [4] (54%), causing increased comorbidities and risk of mortality in this population [3,4,5]. In Spain, the impact of the pandemic represented a 50% drop in the inclusion of patients in 2020 compared to 2019 [4]. Pulmonary tuberculosis can be prevented and cured. About 85% of people who contract it progress satisfactorily with a therapeutic regimen of 4 to 6 months [11]. In addition, treatment reduces transmission. For this reason, it is important to make a quick and correct diagnosis of this pathology [12].

The public health and social measures implemented during the SARSCoV-2 pandemic, such as social distancing and respiratory isolation, may have had a beneficial effect on the transmission of certain infectious agents [13]. A significant decrease in transmission has been demonstrated in infections with short incubation periods, such as influenza or respiratory syncytial virus [14,15]. In pulmonary TB, the precise impact of these measures has still not been determined [16,17]. Furthermore, various studies have stated that the coinfection of SARSCoV-2 and tuberculosis disease could lead to an increase in the severity of COVID-19 and accelerate the progression of TB [15,16,17].

Our working hypothesis is that by leveraging artificial intelligence methodologies on large-scale patient datasets, we can attain a more nuanced understanding of these interrelationships within the context of real-world clinical practice. Big data applications in the health sector, and specifically the application of new technologies to manage and extract value from complex data generated in large volumes of electronic health records, are a reality. Most of the information contained in the medical electronic files is found in an unstructured way, as free text, its analysis being possible through analysis techniques, like artificial intelligence.

The aim of this real-life study is to analyze the epidemiological situation of pulmonary tuberculosis and SARS-CoV-2 coinfection during the first year of the pandemic and to compare the current situation to that existing before the appearance of COVID-19 using artificial intelligence techniques, specifically natural language processing with the Savana Manager 3.0 platform.

## 2. Materials and Methods

This study was conducted in the autonomous community of Castilla-La Mancha, Spain, from January 2015 to December 2020. It is a retrospective observational study designed to follow the guidelines of the Strengthening the Reporting of Observational Studies in Epidemiology (STROBE) Statement.

Data on pulmonary tuberculosis were collected from 2015 to 2020, while data for TB in association with COVID-19 were collected from January 2020 to December 2020. The total population was 2,866,188, and all information was collected from the electronic health records (EHRs) of patients diagnosed with pulmonary tuberculosis and COVID-19.

The data were analyzed using the Savana Manager 3.0 natural language processor, which uses artificial intelligence and big data techniques. This processor is able to analyze unstructured information in EHRs made available by the Castilla-La Mancha Health Service (SESCAM), and then the data are extracted for use in research. A detailed description of the system has been previously published [18,19,20,21]. In addition, through the use of computational linguistic techniques, comprehensive clinical content is scientifically detected and validated using the SNOMED CT tool [22]. This international database uses comprehensive, multilingual, and codified clinical terminology. This concept carries a clinical idea associated with a unique identifier, which is permanent and unalterable. In this way, it attempts to solve the problem of semantic interoperability presented by some classifications such as ICD 10–11, and at the same time, to create an agile mapping with these commonly used classifications. The terms proposed by the SNOMED coding will be used for both COVID-19 and pulmonary TB.

Using EHRead technology, the free text contained in the EHR was analyzed and processed with natural language processing (NLP) techniques. Medical concepts were detected by using computational linguistic techniques and comprehensive clinical content. These unstructured data were treated as big data. We have previously evaluated the performance of Savana to verify the accuracy of the system to identify records that contain mentions of TB, COVID-19, and related variables. The lack of coded clinical data in Spain requires the development of an annotated corpus known as the ‘gold standard’ to carry out this evaluation. This gold standard consists of a set of clinical documents where the appearance of entities/concepts related to pulmonary TB and COVID-19 are manually verified by experts. Specifically, the corpus used in this evaluation is a set of 450 documents reviewed by three experts to guarantee the reliability of the manual annotation and review. Subsequently, a judge external to the study confirms the correct verification of the reviewers. Savana’s performance was automatically calculated using the expert-created ‘gold standard’ as an evaluation resource. This means that the precision of Savana to identify records, in which the presence of the pathology under study and the related variables have been detected, was measured compared to the ‘gold standard’. The evaluation of the system was calculated in terms of the standard metrics of precision (P), recall (R), and its F-score.

Precision (P) = tp/(tp + fp). This parameter offers an indicator of reliability with which the system retrieves the information.

Range (R) = tp/(tp + fn). This parameter offers an indicator of the quantity of information that the system retrieves.

F-measure = (2xprecisionxrecall)/(precisionxrecall). This parameter offers an indicator of the overall data retrieval performance.

In all cases, a true positive (*tp*) was defined as a correctly identified record, a false positive (*fp*) as a wrongly identified record, and a false negative (*fn*) as a record that should have been identified.

Our results regarding Savana’s performance for evaluating pulmonary TB and COVID-19 information are shown in Table 1.

The data collection system complies with the General Data Protection Regulation (GDPR) of the European Union, and it is impossible to identify doctor or patient information when extracting data.

This study has followed all local regulations, procedures for the correct use of big data, Guidelines for Good Pharmacoepidemiology Practices, and the latest edition of the Declaration of Helsinki. The study was approved by the Research Ethics Committee of the Guadalajara healthcare administration (CEIm: 2022.28.EO; acceptance date 19 December 2022). As it is a retrospective observational study that uses patient data anonymously, patient informed consent was not required.

The statistical analysis that was carried out in this study was a descriptive analysis of all the variables evaluated. Absolute frequencies and percentages are expressed as qualitative variables, and means, 95%CI, and standard deviations are expressed as quantitative variables. Student’s *t*-test of the independent samples was used for the analysis of numerical variables. The chi-squared test was used to measure the association and compare proportions between qualitative variables. Differences where the *p*-value was less than 0.05 were considered significant. OpenEpi (v3.0) and SSPS (v25; IBM Corporation, Armonk, NY, USA) were used for the statistical analysis.

## 3. Results

During the study period (January 2015–December 2020), a total of 2592 patients were diagnosed with pulmonary tuberculosis of a total population of 2,866,188 subjects. The flowchart for the study population is shown in Figure 1.

In total, 64.6% of patients diagnosed with pulmonary tuberculosis were men with a mean age of 53.5 years (95%CI 53.0–54.0), and 35.4% were women with a mean age of 52.3 years (95%CI 51.4–53.2).

From January 2020 to December 2020, a total of 210,164 (11.1%) patients were diagnosed with COVID-19. That same year, a total of 438 (0.02%) patients were diagnosed with pulmonary TB, 59.5% of which were male with a mean age of 55.0 years (95%CI 53.4–56.7), and 40.5% were female with a mean age of 53.9 years (95%CI 52.6–55.2). Figure 2 shows the evolution of TB diagnoses annually since 2015. Above the columns, the number of patients diagnosed with pulmonary TB per 100,000 inhabitants is shown. A higher incidence of pulmonary TB was not observed in the first year of the pandemic; instead, a 28% reduction was observed compared to 2019.

Table 2 compares the pulmonary TB diagnoses of 2019 versus 2020, as well as the percentage reduction in diagnoses from one year to the next. A comparative variability between the 2 years stands out (marked in the gray rows of the table), which is probably related to the first and second waves of the COVID-19 pandemic in our setting (March to May, and October to November).

A total of 62 (14.2%) patients diagnosed with TB presented COVID-19 coinfection over the course of 2020, with a mean age of 52.3 years (95%CI 48.3–56.2). In this instance, women were more frequently affected (53.4%; OR 1.8, 95%CI 1.1–3.2). No statistically significant age differences were found between patients with coinfection versus those with a diagnosis of pulmonary TB alone (*p* = 0.18).

The main symptoms of patients with coinfection were dyspnea (27.4%), cough (35.5%), and fever (27.4%). The symptoms of patients diagnosed with TB alone were similar (32%, 36%, and 20%, respectively). Table 3 shows the main comorbidities of patients with coinfection versus patients with pulmonary TB alone. The only statistically significant difference found was for arterial hypertension (*p* = 0.02).

Among patients diagnosed with TB and COVID-19 in 2020, no deaths were recorded from this cause during the study period.

## 4. Discussion

The integration of artificial intelligence (AI) systems plays a pivotal role in assessing the repercussions of the COVID-19 pandemic on the diagnosis and monitoring of various pathologies. In the realm of medicine, AI has emerged as a groundbreaking scientific frontier, orchestrating a paradigm shift in healthcare and biomedical research. Its presence in our everyday clinical practice holds the promise of enhancing the diagnosis, prognosis, and treatment of respiratory diseases. This transformative potential is particularly evident in the application of big data techniques, where AI in healthcare facilitates the management and extraction of valuable insights from the vast and intricate data archived in electronic health records. One of the notable strengths of AI lies in its capacity to handle extensive datasets and discern intricate patterns, fundamentally altering our approach to understanding and managing respiratory diseases. The integration of AI technologies allows for a comprehensive evaluation of the key indicators within specific clinical processes. Crucially, this approach mitigates selection biases, transcending the limitations imposed solely by the existence of a registry.

In the context of the COVID-19 pandemic, where the volume and complexity of medical data have surged, AI systems offer unparalleled advantages. These systems contribute significantly to the rapid and accurate diagnosis of respiratory conditions, enabling timely interventions and personalized treatment strategies. Moreover, AI’s capabilities extend beyond diagnosis, encompassing prognosis and treatment planning. The technology’s ability to sift through massive datasets facilitates the identification of subtle patterns and correlations, potentially unveiling new insights into the progression and management of respiratory diseases.

As we navigate the complexities of the modern healthcare landscape, the symbiosis of AI and medicine stands as a beacon of progress. It not only expedites processes but also ensures a more nuanced and individualized approach to patient care. The ongoing evolution of AI in the medical domain holds vast potential, fostering a future where technology augments our understanding and management of respiratory diseases, contributing to improved patient outcomes and shaping a more resilient healthcare ecosystem.

The aim of this study was to demonstrate this impact on the population diagnosed with pulmonary tuberculosis in 2020 using NLP. The prevalence of this pathology in our setting has been confirmed, which predominantly affected males from 2015 to 2020. This higher prevalence in men has already been described in recent previous studies [11,12].

In 2020, we observed a drop in the incidence of pulmonary TB compared to previous years (shown in Figure 2), including a marked decrease in patients diagnosed in the months of February to April and in October, which was probably due to the first and second waves of COVID-19 in Spain. These results are in line with previous studies from other regions of the Iberian Peninsula and other countries [3,4,5,23,24,25,26,27,28]. The majority of studies previously carried out are of the Italian population, since within Europe it was the country that most quickly suffered the direct causes of the pandemic. These studies present populations studied with a low number of patients, our study being the first to analyze with artificial intelligence the epidemiological situation of pulmonary TB during the pandemic and its possible co-infection with COVID-19.

Several possible explanations exist for this decrease in the incidence of pulmonary TB, which are not necessarily mutually exclusive. The reorganization of hospital services and community health centers due to the pressure of the COVID-19 pandemic could have indirectly affected the management and identification of patients with TB, while also reducing the diagnostic and treatment capabilities for this pathology. Second, it may have caused an underreporting of cases, increasing the number of patients with undiagnosed and untreated pulmonary tuberculosis. Also, the fear of COVID-19 infection or the presence of minor symptoms could have deterred patients from going to medical centers. The same reasons may justify the increase in the proportion of cases who are lost to follow-up.

Another cause of the lower incidence of pulmonary TB during the first year of the pandemic could be the public health and social measures that were implemented. Quarantine, the mandatory use of masks, and social distancing may have had a beneficial effect on certain diseases, resulting in the lower propagation of pathogens spread by airborne transmission, such as pulmonary tuberculosis. In this way, COVID-19 not only aggravates the disaster of this disease, as previously described, but also provides experience in the fight against tuberculosis. People from all social sectors have learned and practiced intervention programs related to respiratory infectious diseases, giving the opportunity for better management of these diseases in the future, raising awareness among the population about taking safety measures.

Coinfection with tuberculosis (TB) and COVID-19 presents a significant medical challenge due to the intersection of two respiratory pathologies with a great global impact. Coinfection can lead to a complex interaction between the two diseases, potentially exacerbating severity and complicating prognosis. The presence of TB can compromise the host’s immune response, increasing the susceptibility and severity of COVID-19 [17]. At the same time, SARS-CoV-2 infection could influence the clinical course of TB, especially in individuals with compromised immune systems. The COVID-19 pandemic has impacted TB control programs, with disruptions to healthcare services and the reallocation of resources. These factors have contributed to delays in the diagnosis and treatment of TB, increasing the risks of transmission and drug resistance. Current research focuses on understanding the mechanisms of interaction between both infections [15,16,17], as well as evaluating the impact of coinfection on progression and clinical outcome. These studies have described a probable increased risk of disease severity in patients with COVID-19 and pulmonary tuberculosis coinfection [15,16,17], which would lead to accelerated progression and symptoms of the latter. With these data documented in the previous bibliography, it is reasonable to speculate that the hyperinflammatory environment induced by COVID-19 could accelerate the progression of TB disease and vice versa. A worrying importance of coinfection with active TB, in addition to the worst results of the treatment, is the possibility of missing a TB diagnosis due to overlapping clinical features.

Strategies are being explored to optimize clinical management and coordination between TB and COVID-19 programs, recognizing the importance of a comprehensive approach that addresses the complexities of these co-occurring respiratory diseases [25,26,27]. Preventing and effectively managing coinfection require close collaboration among health professionals, as well as continued efforts to ensure equitable access to appropriate health services and treatments. A deep understanding of TB-COVID-19 co-infection is essential to guide public health policies and clinical strategies that minimize the impact of this duality of respiratory diseases globally.

In our study population, a total of 62 patients presented coinfection. This study has one of the largest cohorts (both European and worldwide) of pulmonary TB and COVID-19. The female sex was most often affected, as opposed to males in the case of isolated pulmonary TB, and this difference was statistically significant. The increased risk of co-infection in women raises important questions about the interaction between tuberculosis and SARS-CoV-2, as well as potential gender disparities in immune response. Further research is needed to fully understand these findings and determine whether there are biological, sociodemographic, or other factors that explain this association. There were no major changes with respect to age when diagnosed with pulmonary TB alone, remaining at around 50 years of age. The symptoms presented by coinfected patients were similar to patients who presented TB alone, which were dyspnea, cough, and fever. The comorbidities of these patients were also studied, although patients with COVID-19 and TB did not present more comorbidities. This information could have important implications for clinical management and public health strategies, as it suggests that people with co-infection do not necessarily experience a more severe course of the disease. However, continued surveillance and more extensive research is needed to fully understand the clinical and epidemiological ramifications of this dual interaction between pulmonary tuberculosis and COVID-19. In this group, no related deaths were registered.

## 5. Conclusions

Despite the emergence of the COVID-19 pandemic, we have not observed a rebound in the incidence of pulmonary tuberculosis. This may be due to the delay in the diagnosis of the disease in the first wave or the public health and social measures adopted in that timeframe. In our setting, women presented an increased risk of pulmonary TB and SARS-CoV-2 coinfection. These patients did not present more comorbidities or worsened symptoms compared to patients with isolated TB diagnosis. This finding suggests a unique dynamic between tuberculosis and COVID-19, where specific factors, yet to be determined, could influence women’s susceptibility to this co-infection.

## Figures and Tables

**Figure 1 jpm-13-01629-f001:**
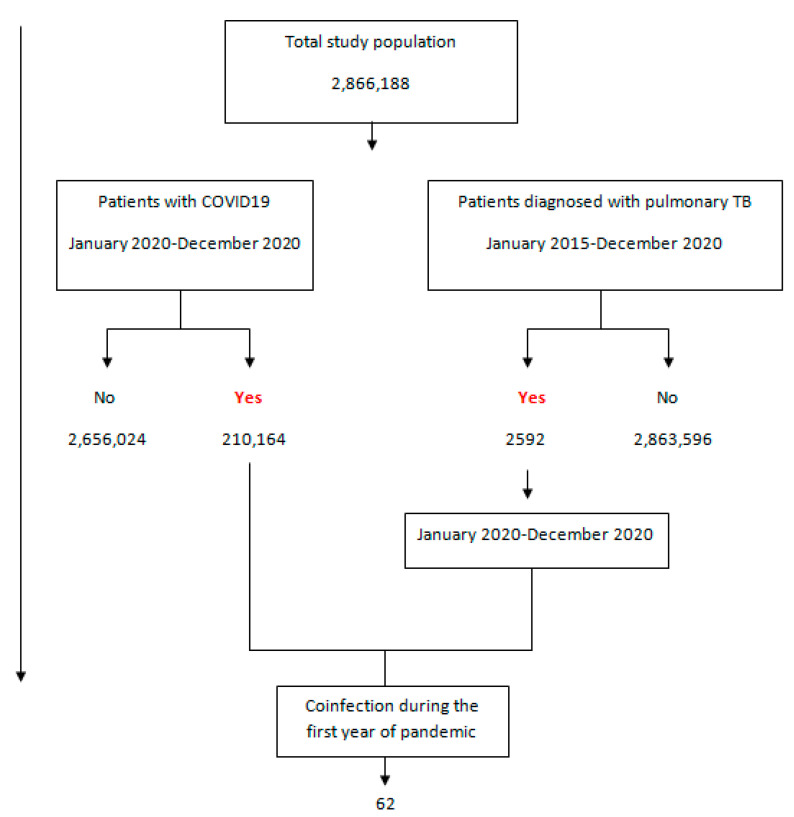
Flowchart showing the total study population, patients diagnosed with COVID-19, patients diagnosed with pulmonary TB, as well as patients with coinfection of these two diseases during the first year of the pandemic.

**Figure 2 jpm-13-01629-f002:**
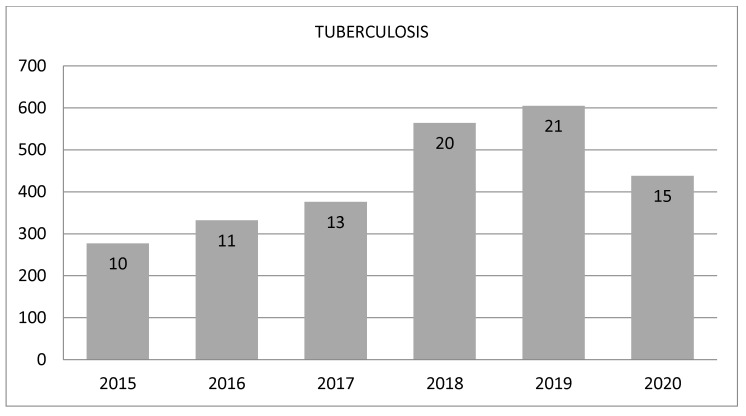
Annual prevalence of pulmonary TB from 2015 to 2020. Within the columns, TB pulmonary patients per 100,000 inhabitants are shown.

**Table 1 jpm-13-01629-t001:** Performance of Savana in terms of precision, recall, and F-measure for IPF and COVID 19.

	Precision	Recall	F-Measure
PULMONARY TB	1.0	0.94	0.97
COVID-19	0.99	0.75	0.93

**Table 2 jpm-13-01629-t002:** Comparison of pulmonary TB diagnosis in 2019 and 2020.

	Pulmonary Tuberculosis		
	2019	2020	Reduction
January	62	59	4.84%
February	50	40	20.00%
March	58	33	43.10%
April	60	10	83.33%
May	69	30	56.52%
June	43	41	4.65%
July	49	45	8.16%
August	23	21	8.70%
September	34	31	8.82%
October	70	45	35.71%
November	52	50	3.85%
December	35	33	5.71%

**Table 3 jpm-13-01629-t003:** Main comorbidities in patients with TB + COVID-19 coinfection versus patients with pulmonary TB alone.

	Pulmonary TB +COVID	%	Pulmonary TB	%	OR (IC 95%)
Arterial hypertension	21	33.9	200	45.7	0.6 (0.3–0.9)
Diabetes mellitus	15	24.2	136	31.1	0.7 (0.4–1.3)
Dyslipidemia	20	32.3	160	36.5	0.8 (0.5–1.4)
Obesity	6	9.7	45	10.3	0.9 (0.4–2.3)
COPD	5	8.1	58	13.2	0.5 (0.2–1.4)
Asthma	4	6.4	29	6.6	0.9 (0.3–2.8)
Ischemic cardiopathy	2	3.2	24	5.5	0.5 (0.1–2.3)
Human immunodeficiency virus (HIV)	3	4.8	44	10.0	0.4 (0.1–1.4)

## Data Availability

The data presented in this study are available upon request from the corresponding author. The data are not publicly available because they belong to the national health system of Castilla La Mancha.

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
