# Peer review of "Impact of the COVID-19 Pandemic on the Epidemiological Situation of Pulmonary Tuberculosis–Using Natural Language Processing"

_jpm, 2023, doi:10.3390/jpm13121629_

Round 1
Reviewer 1 Report
Comments and Suggestions for Authors
This article raises an important and relevant topic about the diagnosis of TB in the context of the Covid-19 epidemic. The use of artificial intelligence to analyze the influence of various factors on morbidity, in this case the influence of Covid on TB, seems extremely promising. Could be given the exact numbers of incidence per 100,000 above the columns in Figure 2? And also absolute numbers for percentages in lines 185 - 187?
Author Response
This article raises an important and relevant topic about the diagnosis of TB in the context of the Covid-19 epidemic. The use of artificial intelligence to analyze the influence of various factors on morbidity, in this case the influence of Covid on TB, seems extremely promising. Could be given the exact numbers of incidence per 100,000 above the columns in Figure 2? And also absolute numbers for percentages in lines 185 - 187?
Thank you for your comment. Changes have been added to the document.
Reviewer 2 Report
Comments and Suggestions for Authors
Covid-19 has had a negative impact on the global control of tuberculosis. Delay in diagnosis, patient follow-up and treatment compliance problems have increased.
However, covid-19 did not lead to an increase in mortality in patients with tuberculosis.
It is a well-planned study that reveals this situation. But what additional benefit has AI provided over normal statistical analysis? These results can already be determined with basic descriptive statistics methods.
Author Response
Covid-19 has had a negative impact on the global control of tuberculosis. Delay in diagnosis, patient follow-up and treatment compliance problems have increased.
However, covid-19 did not lead to an increase in mortality in patients with tuberculosis.
It is a well-planned study that reveals this situation. But what additional benefit has AI provided over normal statistical analysis? These results can already be determined with basic descriptive statistics methods.
The advancement of new technologies and the possibility of performing massive data analysis allow us to know the situation of pulmonary TB in situations real life that, due to biases of different kinds, cannot always be correctly assessed with other methodologies.
This study identifies what are the main characteristics of a population unselected TB with COVID19 coinfection. This information allows you to guide care strategies that are effective and that help improve the situation of this disease in our environment, all with the possibility that the effect of such measures can be monitored in a continuous form.
Reviewer 3 Report
Comments and Suggestions for Authors
Some paragraphs contain citations only at the very end (e.g. lines 46-51), but very different information. It would be better if the specific source were mentioned after each piece of information.
Methods: "artificial intelligence and Big Data techniques" - please specify, which ones where used
"In addition, through the use of computational linguistic techniques, comprehensive clinical content is scientifically detected and validated by the SNOMED CT tool" - please explain more in detail what was done here.
Line 110-112 - please cite your previous evaluation
Line 107 - 137, including table 1: I would see this more as a result or part of the supplement. It doesn't fit really well here.
Line 123: Does the 19 have a meaning?
table 2: Please make it clear what the gray table rows are supposed to mean.
Please write the P in lower case for the p-values
Discussion:
I don't quite understand the conclusion - it is conveyed in the discussion that diagnosis can be improved based on technology. But this is not shown in the study. The study is exclusively descriptive based on an AI-generated data set and does not show any diagnostic approaches.
Comments on the Quality of English LanguageSome sentences are unnecessarily long and could be shortened (for example: "The tubercle bacillus (Mycobacterium tuberculosis) is the second..." - you could just say "M. tuberculosis is the second...")
Author Response
Some paragraphs contain citations only at the very end (e.g. lines 46-51), but very different information. It would be better if the specific source were mentioned after each piece of information.
Changes have been made to the manuscript.
Methods: "artificial intelligence and Big Data techniques" - please specify, which ones where used
As mentioned in the article, through technology EHRead (included in the field of artificial intelligence), the free text contained in the EHR was analyzed and processed with NLP. This unstructured data has been treated as big data.
"In addition, through the use of computational linguistic techniques, comprehensive clinical content is scientifically detected and validated by the SNOMED CT tool" - please explain more in detail what was done here.
We have added an explanation to material and methods.
Line 110-112 - please cite your previous evaluation
This evaluation consists, as explained in the article, in the manual analysis of a set of clinical documents by two independent reviewers. This validation has been carried out on a total of 450 documents. Subsequently, a judge external to the study confirms the correct verification of the reviewers.
Line 107 - 137, including table 1: I would see this more as a result or part of the supplement. It doesn't fit really well here.
Thanks to the reviewer for his comment, we still believe that this part is essential for the description of material and methods, and especially to know the gold standard for the evaluation of the analyzed terms. The editor has recommended to us to expand the description of the methodology, which is also why we have kept it in the article. For this reason we believe that this part of the article should not be changed.
Line 123: Does the 19 have a meaning?
The bug has been fixed. Thank you.
table 2: Please make it clear what the gray table rows are supposed to mean.
A brief comment about this has been added to the article.
Please write the P in lower case for the p-values
It has been corrected in the document in this section. Thank you.
Discussion:
I don't quite understand the conclusion - it is conveyed in the discussion that diagnosis can be improved based on technology. But this is not shown in the study. The study is exclusively descriptive based on an AI-generated data set and does not show any diagnostic approaches.
Our study is not intended to create a diagnostic method. As described in the discussion, our work aims to analyze covid19 and pulmonary TB coinfection in a real-life situation using artificial intelligence. By analyzing this large amount of data from more than 2866188 patients, we aim to avoid selection biases that are present in conventional patient registries
Comments on the Quality of English Language
Some sentences are unnecessarily long and could be shortened (for example: "The tubercle bacillus (Mycobacterium tuberculosis) is the second..." - you could just say "M. tuberculosis is the second...")
Thank you for your comment, changes have been made to the article.